# Virtual histological staining of tissue from a single label-free autofluorescence image using deep learning

**Yair Rivenson**[*1]                                   RIVENSNONYAIR@UCLA.EDU
[1] *UCLA Department of Electrical and Computer Engineering*

**Hongda Wang**[*1]                                   HDWANG@UCLA.EDU
**Zhensong Wei**[1]                                   ZWEI@UCLA.EDU
**Kevin de Haan**[1]                                   KDEHAAN@UCLA.EDU
**W. Dean Wallace**[2]                          WWALLACE@MEDNET.UCLA.EDU
[2] *UCLA Department of Pathology and Laboratory Medicine, David Geffen School of Medicine*

**Aydogan Ozcan**[1]                                   OZCAN@UCLA.EDU

**Editors:** Under Review for MIDL 2019

## Abstract

Histochemical staining of tissue samples is required for the diagnosis of many diseases, including cancer; however, the staining process is often time consuming, slow, costly, and does not support tissue preservation for advanced molecular analysis of the sample. Recently, we presented a deep learning framework that can perform virtual histochemical staining, in silico, of unlabeled tissue sections using a single autofluorescence image, emanating from the tissues endogenous fluorophores(Rivenson et al., 2019b). We validated the success of this technique through a direct comparison between the virtually and histochemically stained slides, as well as by a blind study performed by a panel of board-certified pathologists.

**Keywords:** Deep learning, Histology, Digital pathology

## 1. Introduction

A wide variety of diseases are diagnosed through histological analysis. It is typically done through a long and labor-intensive process which involves sectioning a tissue sample into a thin slice, mounting it to a microscope slide, and labeling it chemically. Following these steps, the labeled tissue section can be microscopically studied by pathologist using a brightfield microscope. In recent years there have been several attempts to change this process and eliminate the need for chemical labeling. These attempts have used methods such as Raman scattering(Ji et al., 2013) second harmonic generation(Tao et al., 2014), or a combination of different imaging modalities(Tu et al., 2016). Furthermore, some of these imaging modalities have been augmented with pseudo-hematoxylin and eosin (HE) staining, to accommodate for the diagnostic gold-standard(Tao et al., 2014), (Giacomelli et al., 2016). However, most of these methodologies scan slowly and require high-end equipment. Pseudo-staining procedures were limited to HE, with simplified approximations to the dye concentration and studies mostly limited to one type of tissue. Recently(Rivenson et al., 2019b), we demonstrated that using a single autofluorescence image of a tissue section, captured with a standard fluorescence microscope, we can create a virtually-stained image

---

[*] Contributed equally

using deep learning, matching the brightfield image of the histologically stained version of the same tissue sample. We demonstrated the efficacy of this approach with various tissue and stain combinations that were examined by a panel of board-certified pathologist.

## 2. Methods

For each tissue and stain type, an image dataset made up of co-registered autofluorescence images of label-free tissue samples and the corresponding brightfield histochemically stained tissue images were created. The unlabeled tissue slides were first imaged using the autofluorescence emission from near-UV excitation and then histochemically stained and imaged again using a brightfield microscope. Following that, the images were registered using a series of steps beginning with a coarse matching of the fields of view, determined by cropping out the area with the highest correlation and then correcting for size and rotation. To finely register the images, a network was first trained to perform a rough transformation between the images. An elastic registration algorithm was then used to match the histochemical stained labels with the output of the network at a subpixel level.

In order to generate the virtually stained images of label-free tissue samples, a generative adversarial network (GAN)(Goodfellow et al., 2014) was used in conjunction with an L1 and total variation loss(Rudin et al., 1992) as regulaizers. A U-net architecture was used as the generator(Ronneberger et al., 2015), while a VGG style classifier(Simonyan and Zisserman, 2015) was used as the discriminator. A diagram of the network structures can be seen in Figure 1.

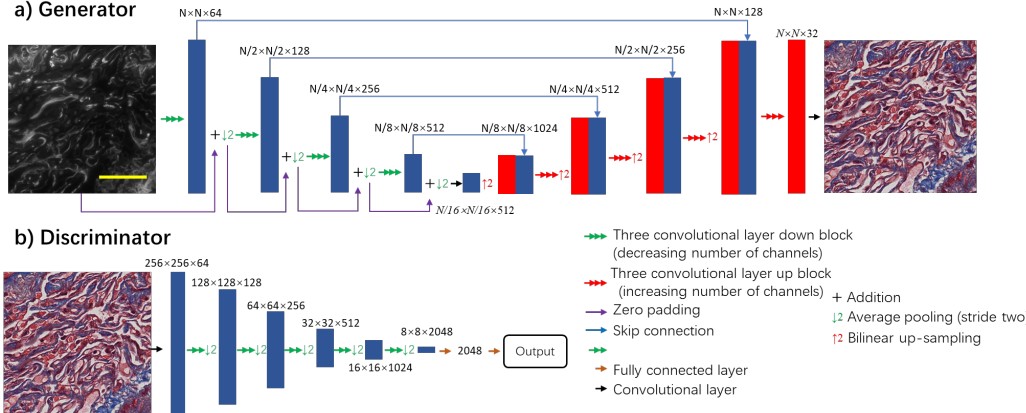

Figure 1: Diagram of the network structure. a) Diagram of the generator network. b) Diagram of the discriminator network. Scale bar indicates 50 µm.

## 3. Results

We performed the virtual staining using our approach with a variety of tissue and stain combinations. Three examples of these can be seen in Figure 2, which demonstrates a comparison between the networks output and the ground truth histochemically stained tissue samples. This figure demonstrates the networks ability to accurately transform three different tissue types (salivary gland, liver, kidney) and three different stains (HE, Massonss

trichrome, Jones stain). Other tissue types such as lung, prostate, and skin were also used for diagnostic validation of our images(Rivenson et al., 2019b).

In addition to direct comparisons between the images, our technique was further validated in a diagnostic setting by a panel of four board-certified pathologists. During a blind test, the pathologists were given 15 tissues to diagnoses. This analysis indicated that there were no clinically significant difference between the diagnoses made with virtually stained images vs. the standard histochemically stained ones(Rivenson et al., 2019b). An additional blind study performed by 6 pathologists was used to judge the quality of the staining technique. These pathologists were given a series of virtually stained whole slide images (WSIs) and the corresponding histochemically stained images. They rated different aspects of the staining quality for both slides, and no clear preference was found for either staining technique(Rivenson et al., 2019b).

An added benefit to virtually staining tissues is that the process is more repeatable than manually prepared histochemical staining. This lack of variation can be helpful for both pathologists and any software used for e.g., automated analysis and classification.

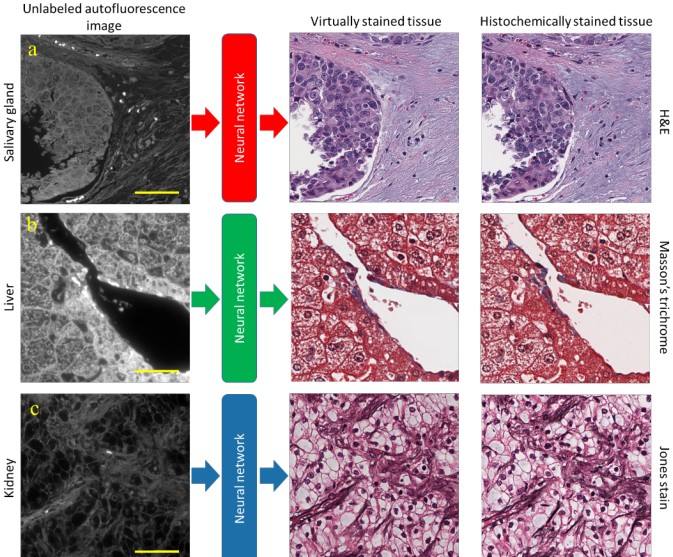

Figure 2: Demonstration of deep learning-based virtual staining using three different stains and tissue types. a) salivary gland tissue, b) liver tissue, c) kidney tissue. Scale bar indicates 50 µm.

## 4. Conclusions

This presented technique has the potential to transform histopathology workflow by eliminating the need for histochemical staining of tissues, which will reduce costs, enable histotechnologists to perform more advanced analysis on the preserved tissue and allow overall faster diagnoses. The technique can also benefit from additional wavelengths of autofluorescence(Rivenson et al., 2019b) and can be generalized to other imaging modalities, such as phase imaging(Rivenson et al., 2019a). Furthermore, the staining standardization can foster more accurate and reliable diagnosis for both human experts and machine learning tools(Liu et al., 2017).

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
