# OpenReview forum: "Virtual histological staining of tissue from a single label-free autofluorescence image using deep learning"
_MIDL.io/2019/Conference/Abstract — MIDL Abstract 2019_

### Official Review · AnonReviewer1 · 2019-04-29
**Interesting abstract even though GANs can hallucinate features and therefore convey false information**

**Rating:** 3
**Confidence:** 2

**Review:**

The abstract reads very well and I believe that, given the length constraints of the abstract format, the authors have fit sufficient information about the method in order to both describe and ensure reproducibility of their approach.

The method is technically sound. A encoder-decoder architectures is used together with an adversarial loss to virtually stain microscopy images and change their colorization so that they match the distribution of true stained slides. The results look visually impressive and the fact that authors try their technique on different kinds of data, meaning different kinds of cells from different tissue, is somehow reducing the concerns that one would usually have when using GANs to convert an image type into another.

The quantitative results achieved by this method aren't discussed in depth. It would be nice to see what are the performances of this approach in terms of more quantitative metrics.

Another concern, which is my main concern, is related to the fact that GAN can hallucinate features in medical image translation. As the work of JP Cohen "Distribution Matching Losses Can Hallucinate Features in Medical Image Translation" demonstrates, the practice of using GANs to accomplish tasks similar to the one shown in this paper, is dangerous.

The authors need to clarify if the autofluorescence images contain the same information that the stained images contain, which means that the network should not be given the task to guess information that is not contained in the raw data using some "mysterious cues" and some "magical procedures". The question therefore is: "is the information that we observed in stained images already present in autofluorescence images?". Unfortunately, since microscopy is not my main field, I am unable to answer the question myself (this is the reason why I am going for confidence level 2 out of 3 for this review).

This work is not particularly novel, but I found it interesting and well explained in very compact format.

---

### Official Review · AnonReviewer2 · 2019-04-29
**Abstract Review**

**Rating:** 3
**Confidence:** 2

**Review:**

The authors proposed a Generative Adversarial (GAN) architecture for virtual histological staining in a recently accepted journal paper (Rivenson et al., 2019b) and the abstract summarizes the findings. A good agreement between virtually and histochemically stained slides have been reported in accordance to a blind study performed by a panel of pathologists.

Pros

- A thorough study on the utility of GANs in virtual histological staining.
- The proposed method is well explained.
- Highly confident results thanks to the analysis if board-certified pathologists.

Cons

- No quantitative results are given to analyze the final image quality (e.g. MSE, PSNR).
- The proposed GAN is a state-of-the-art and there is a significant literature on the GANs for medical image analysis (Kazeminia et al., GANs for Medical Image Analysis, arxiv 2019). The authors do not have any method of comparison for performing the same task (e.g. U-net, cycle-GANs) to truly investigate the value of the proposed technique.

Minor issues
- Registration process can be better explained. For example the elastic registration algorithm is not cited and the citation can be useful for reproducibility.

---

### Decision · Program_Chairs · 2019-05-06
**Acceptance Decision**

Accept